# Are Small Agricultural Markets Recipients of World Prices? The Case of Poland

Anna Szczepańska-Przekota 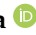

Faculty of Economic Science, Koszalin University of Technology, Kwiatkowskiego 6E, 75-343 Koszalin, Poland; anna.szczepanska-przekota@tu.koszalin.pl

**Abstract:** The increased inflation in 2021–2022, and in particular the increase in the prices of energy carriers, and thus chemical fertilizers, caused an imbalance in the market of agricultural raw materials in Poland. This problem, to a greater or lesser extent, can also be observed in other countries. Meanwhile, the issue of shaping domestic prices of agricultural commodities is one of the most important problems in a state's food policy. This is evident in countries with strong agricultural traditions, such as Poland. Many tensions and misunderstandings between agricultural producers and the government concern the low purchase prices of agricultural commodities. Therefore, the degree of integration of the Polish price market with the world market was studied. Based on data from the wheat, beef and pork livestock markets, the impact of the world market on the Polish one was studied using VAR methodology. The analyzed data concern the years 2012–2022. It was found that the degree of price integration of various agricultural commodities is different, but always positive. The market of wheat turned out to be the most strongly price-integrated, and the markets of beef and pork livestock are slightly weaker. Such results call into question the effectiveness of aid programs for agricultural producers, as there will always be costs for the budget, which will have to be paid by the next generations, will not cause the increase in commodity prices expected by agricultural producers and will also be contrary to the principles of the free market.

**Keywords:** agricultural commodities; integration; market; causality; prices

## 1. Introduction

The national economy of each country consists of many interrelated subsystems. Due to factors such as geographical location, availability of natural resources and level of technological advancement, each country is characterized by a different structure of the national economy. In some countries, the dominant sector may be the mining industry, in others it may be the service industry and in others it may be the agricultural sector. However, regardless of the structure of the national economy, each country tries to provide its citizens with the opportunity to meet the basic needs of life, and the most important of these are nutritional needs [1]. In countries such as Poland, the agricultural sector and, more broadly, the food sector are traditional sectors that evoke a lot of social emotions, especially in times of perturbation and market imbalance. In many respects, Poland could be nutritionally self-sufficient, which is why farmers are very reluctant to accept competition from other countries and demand for protective measures, including tariffs and bans. How the transmission from world prices to domestic prices takes place and what the scale of this phenomenon is are key problems in determining the tasks of state policy and an indicator of the effectiveness of market interventions [2,3].

The demand for agricultural commodities is more evenly distributed over time than supply [4]. This causes periodic surpluses and shortages and requires a development in warehouse management [5]. According to the laws of the market, the existing imbalance should be balanced by prices. On the supply side of agricultural products, three main factors are assumed to be responsible for price changes: weather phenomena, a slowdown

in grain production growth and rising oil prices [6,7]. On the demand side, two of the most important drivers are population growth and increasing wealth. These cause quantitative and qualitative changes in the demand for food [8–10]. In Poland, as in many other European countries, especially in Central and Eastern Europe, a rapid increase in wealth can be observed. Although the COVID-19 pandemic, lockdown, inflation and the influx of the Ukrainian population to Poland have adversely affected the income of Polish society, from a long-term perspective, the changes are definitely positive. The expectations of the population, diet, health awareness and, as a result, the demand for agri-food products have changed [11]. There is no doubt, however, that agricultural commodities such as grains and livestock will always be the basis of human nutrition. What is characteristic of plant production, and to a lesser extent animal production, is seasonality.

Seasonal variability accompanies many economic phenomena. It can be defined as a systematic movement of a certain quantity during the year. Sources of seasonal fluctuations are varied and generally result from supply and demand conditions changing throughout the year. One well-known example of seasonal fluctuations is changes in the supply and prices of agricultural products. Many of them are characterized by specific production cycles, conditioned by climatic factors. As a consequence, peaks in supply occur during the harvest season [12].

The formation of prices of agricultural commodities is partly the result of the operation of the free market [13]. In periods of increased supply, there is a decrease in prices, and in periods of lower supply, prices increase. In general, demand is more stable than supply, which is natural and results from the nature of agricultural production. Prices are regulated by the market, but this mechanism can be disrupted by institutions [14]. The problem of demand for agricultural products is related to the fact that it is dependent on the consumption of processed food. A rising national income causes changes in the way that people eat, and thus also causes changes in the structure and volume of demand for agricultural products. Only a small part of the production produced on farms becomes the final goods. The vast majority is further processed both by the food industry and by the fuel, textile, paper and other industries. Before reaching the final consumer, agricultural products undergo a variety of transformations (temporal, spatial and material). The processing of agricultural products is associated with additional costs of labor, means of production, other raw materials, etc. In the final product, the agricultural commodity is only a component of the product with different, new utility and/or taste qualities, which is used to meet consumer needs. As a result of additional costs between the agricultural producer and the consumer, there are significant differences in the prices paid by consumers for food products and those received by agricultural producers [15,16]. This phenomenon is referred to as vertical price transmission in marketing traffic. There is no strong relationship between price changes in the vertical arrangement of the marketing chain. As a rule, the prices of final products are more rigid than the prices of agricultural raw materials [17,18].

An important factor affecting demand in the domestic market is the demand for products reported by foreign markets [19–21]. Under normal circumstances, the size of such demand will be determined by the needs of foreign markets and by price relations that take into account exchange rate changes [22,23]. However, as economic practice shows, very often, non-economic or, more precisely, political considerations come into play here. The problem of price relations is quite serious here, because there are many trading companies in the market that take advantage of price differences and, depending on the situation, import or export agricultural products. Particularly controversial is the import of products also manufactured in a given country [24,25].

Price fluctuations in the market of agricultural commodities over the last decade have been a topic of interest for many researchers. Some of them are concerned with the impact of fundamental factors; others indicate that, apart from fundamental factors, such as the specificity of the agricultural market and macroeconomic and financial factors, there are still others. The most common among these other factors is the so-called financialization of agricultural commodity markets [26,27]. It manifests itself in the growing role of financial

motives, other financial markets and financial players in the functioning of agricultural commodity markets [28].

Financial investors have been active in commodity agricultural contract markets since their inception, but in recent years, their activity has increased in particular [29]. This has come from the belief that commodities, including agricultural commodities, as an asset class, provide a hedge against other securities in financial markets. This has encouraged financial investors to expand into agricultural future markets. The belief in the possibility of securing a position in the commodity market has been supported by empirical research, which has shown that the rates of return in the commodity future markets are negatively correlated with the rates of return on the stock and bond markets. This has caused a very negative phenomenon for agricultural markets, namely, many analysts note that, currently, there is no natural correlation between the volume of demand and supply and price in agricultural markets. Thus, the agricultural market appears to be a destabilized market [30–32]. From the point of view of agricultural producers and processors of agricultural commodities, this makes long-term hedging against the risk of changes in prices of agricultural raw materials more costly and complicated [33].

Due to the volatility of weather phenomena, the seasonal cycle of agricultural production and the related volatility of food prices, the risk of agricultural production increases. Under such conditions, proper planning and analysis of various development scenarios become crucial. Given that combinations of key trends and risk factors can lead to different future outcomes, agricultural sector development scenarios broaden the perspective and provide information about the consequences of the choices made today [34]. Risk management in agricultural production is all the more important, as the agricultural sector affects other areas of a given country's economy. Theoretically, future contracts for agricultural products can be used to effectively set prices for the future, plan production, manage risk and minimize the effects of seasonality in production and consumption, which means that the effectiveness of investments in agricultural production can be high [35,36]. These agreements can contribute to reducing uncertainty in the agricultural products market, and they can be an important risk mitigation measure for industries that use agricultural sector products as raw materials. However, such contracts come at a cost, and the prospect of contract mispricing can be far-reaching. Therefore, the integration of domestic cash markets with foreign future markets can be seen in the market.

The activity of modern future markets in the field of providing information for real markets is so important that it is the primary mechanism for shaping spot prices, especially for agricultural products with seasonal production, whereby the sale of which is spread over time. There are known studies on the dependence of spot future prices on the grain market. The results strongly suggest that grain future prices are used as a benchmark for spot prices and that changes in future prices usually lead to changes in spot prices [37,38].

The prices of agricultural products have been quite strongly influenced by energy policy in recent years. The use of biofuels has resulted in an increase in demand for certain agricultural commodities and a stronger link between agricultural production prices and crude oil prices [39]. The impact of this policy manifests itself in two ways—through the relation of changes in oil prices with the prices of plants intended for biofuels, and through changes in the share of biofuels in the final product. Changes in the relationship between crude oil prices and the prices of plants used in the production of biofuels affect the sowing structure, e.g., rising oil prices increase the profitability of, for example, rapeseed cultivation and increase its price, but on the other hand, a larger area of cultivation means a greater supply, which acts as a constraint on prices. However, the net effect is generally positive and the pressure from oil prices is stronger [40]. The opposite will be true in the case of falling oil prices. In addition, incentives in the form of tax exemptions or subsidies may cause greater interest in growing plants for biofuels, even at low oil prices [41].

The lack of stability of risk factors makes it difficult to forecast prices based on fundamental indicators, and sometimes simple models based on seasonality can be more effective. Seasonal fluctuations are characterized by relatively high repeatability. In the agricultural

market, this applies primarily to the supply of grains. It is known that it is always the highest in August, regardless of whether there has been a drought or a lot of rain in a given year. This is a natural consequence of the vegetation cycle. The supply and purchase of grain in August is so large that, practically every year, prices in August are the lowest in the annual cycle. The decision-making problem for the farmer is whether to sell the grain after the harvest or to store it, hoping for a higher price in the coming months. If the choice falls on storage, the following questions arise: what time will be the best for sale, and will the higher price obtained cover the costs of storage [42–44]?

There is no future exchange for agricultural commodities in Poland; therefore, any attempts to hedge cash prices must be made using forward contracts or foreign future contracts. The development in electronic means of communication and the liberalization of capital flows make the use of such solutions easy. The only problem may be the effectiveness of achieving the objectives of strategies based on foreign contracts [45]. The studies conducted so far on hedging cash prices of wheat using future contracts from the MATIF exchange (Euronext Paris) show that cash prices in Poland and contract price quotations are positively correlated, which is due to global trends, but in the short term, local specificities become visible, and quotes may behave differently. From the point of view of the hedging strategy, however, the results were not very optimistic. The different conditions of the Polish and French markets did not allow for the effective use of future contracts. In particular, it was noted that the agricultural producer's income becomes unstable after using forward instruments. This changes the nature of the transaction from a hedging one to a speculative one [46].

The interplay of demand and supply on the internal market could balance prices on a national scale, but these relations depend to a large extent on the economic situation on a global scale. The equilibrium price in the domestic market depends on the supply and demand situation in other countries. Under the conditions of trade liberalization, a shortage in the world market will put pressure on prices in the country due to increased exports, and, vice versa, a surplus in the world market will put pressure on prices in the country due to increased imports. In such a situation, the equilibrium price will be set at the global level. This process is referred to as the law of one price [47]. The effectiveness of this process depends on a number of factors, in particular on transfer costs, state interference in the market, transaction costs and the flow of information between exchange participants. In addition, it is indicated that price flows from a larger economy (larger producer) to a smaller economy (smaller producer) are stronger than flows in the other direction. The indicated sizes of economies together with the distance between them determine the weights for the level of trade and price transmission. In the global economic system, Poland is considered a small country and for this reason it becomes a recipient of the world price.

One of the basic reasons why Poland can be considered as a recipient of world prices is the issue of the exchange rate. Even if we are among the world leaders in the production of some agricultural commodities, changes in the exchange rate still put Polish producers at risk. Trade on the international arena influences the exchange rate; so, it is shaped by movements related to food exports and imports. However, in the overall trade, the share of agricultural raw materials is small; therefore, the agricultural market is forced to adopt the exchange offered by the world market. It follows that changes in the prices of domestic agricultural commodities adjust to changes in world prices.

According to international trade theory, the convergence of domestic prices with world prices contributes to general welfare [48,49]. International trade reduces the risk of price volatility and helps to stabilize the income of agricultural producers. In a situation of surplus and falling domestic prices, they can liquidate the surplus in the domestic market, which balances the market and has a stabilizing effect on the price. However, this phenomenon carries the risk of transferring shocks from the foreign market to the domestic market, for example, in a situation of surplus in the world market and poor yields in the domestic market. If the world price is lower than the domestic price and the domestic market is not protected, imports will increase, which will translate into poorer performance

for domestic producers. Ultimately, however, it is indicated that the main beneficiaries of the trade are consumers, and producers in some situations may be exposed to additional sources of risk.

Taking into account the above observations, two main questions were formulated:

Q1. Is the Polish agricultural market a recipient of prices from the world market?

Q2. Is the price volatility of the domestic market of agricultural raw materials higher or lower than the price volatility of world markets?

When talking about domestic prices, we are referring to buying prices in Poland, while when talking about world prices, we are referring to contract prices amongst some of the world's largest exchanges of CBOT and CME.

The research conducted here fills an important research gap. Determining the degree of integration of the domestic market with the world market is not a new issue, but embedding it within one market, broken down into individual products, is quite rarely performed. Sometimes one talks about the market in general terms and draws general conclusions from it, whereas there may be specific characteristics within the market. In this article, they are shown based on the agricultural market.

## 2. Materials and Methods

The considered data for the empirical analysis are the purchase prices of the most important agricultural commodities in Poland:

1. Wheat (in PLN/q); (q = 0.1 kg).
2. Cattle (in PLN/kg).
3. Hogs (in PLN/kg).

These are the average monthly purchase prices in processing plants recorded by the Central Statistical Office in Poland.

These prices were compared with the original future prices:

1. Wheat CBOT (c/bu); (c = 0.01 USD; bu = 27.216 kg for wheat).
2. Live cattle CME (c/lb); (1 lb = 0.454 kg).
3. Lean hogs CME (c/lb).

The analyzed series are the average monthly quotations of these contracts. After taking into account the USD/PLN exchange rate and converting the bu-q and lb kg metric units, the prices of global contracts are presented in Polish metric units:

1. Wheat CBOT (in PLN/q).
2. Live cattle CME (in PLN/kg).
3. Lean hogs CME (in PLN/kg).

All series are monthly data for 2012–2022. Such series include 132 observations each. This length turned out to be sufficient to find significant relationships.

The research procedure included several stages:

1. Graphical presentation of time series of agricultural commodity prices and their rates of return along with descriptive statistics:

   - Time series of prices—mean, standard deviation, coefficient of variation;
   - Time series of returns—standard deviation, minimum, maximum.

2. A simple correlation study:

   - Analysis of Pearson's linear correlation coefficient $r_{XY}$ between domestic and world agricultural commodity prices;
   - Spearman $R_{XY}$ rank coefficient analysis between domestic and world agricultural commodity prices;
   - Determining the degree of curvilinearity of domestic and world prices of agricultural raw materials:

$$m_{XY} = R_{XY}^2 - r_{XY}^2. \tag{1}$$



3.  Granger causality study between domestic and world prices of agricultural commodities. The idea of Granger causality is based on Equation [50]:

$$y_t = \alpha + \sum_{k=1}^{K} \gamma_k y_{t-k} + \sum_{k=1}^{K} \beta_k x_{t-k} + \varepsilon_k \text{ with } t = 1, \ldots, T \tag{2}$$

where $x_t$ and $y_t$ are two stationary series. Model (1) can then be used to test whether $x$ causes $y$. Essentially, if the past values of $x$ are significant predictors of the current value of $y$, even when the past values of $y$ have been included in the model, then $x$ exerts a causal influence on $y$. Using (1), one might easily investigate this causality based on an $F$ test using the following null hypothesis:

**Hypothesis 0 (H0):** $\beta_1 = \cdots = \beta_K = 0$.

If H0 is rejected, one can conclude that causality from $x$ to $y$ exists. The $x$ and $y$ variables can be interchanged to test for causality in the other direction, and it is possible to observe bidirectional causality (also called feedback) [51].

## 3. Results

### 3.1. Shaping Prices and Rates of Return of Agricultural Commodities

The prices of basic agricultural commodities over the analyzed period of 2012–2022 were subject to significant changes (Figure 1, Table 1). This concerned both the long-term trend direction and the scale of short-term volatility.

Wheat price trends in the Polish market and the world market are consistent and relatively stable, i.e., if the prices enter the trend, it lasts for several years. Throughout 2012, prices increased; then, they were in a downward trend until 2018; then, from 2018 they entered an upward trend and in 2022 they began to fall. The highest prices were recorded in May 2022, and the lowest were recorded in 2016–2018 with the differences between the lowest and highest prices reaching 180%. In Poland, the lowest recorded price was PLN/q 59.54 and the highest was PLN/q 167.36, while on the CBOT exchange the lowest recorded price was 398.69 c/bu and the highest was 1113.69 c/bu. Very large spreads apply to monthly changes: as for the Polish market, they were from −19.5% to 19.9%, and for the US market, they were from −14.4% to 25.0%. In general, the overall volatility of wheat prices (coefficient of variation) and returns (standard deviation) is quite significant. In the long term, the Polish market was less stable than the global market—the price volatility coefficients were 30.6% and 25.4%, respectively—but in the short term, the Polish market was more stable than the global market—the standard deviation of the rates of return was 4.99 pp and 5.89 pp, respectively.

Live cattle prices in the Polish and world markets behaved differently throughout 2021. In the Polish market, they were in a sideways trend with minor corrections of up to 10%, while on the CME exchange, the price differences reached 50%. From 2021, very large price increases are visible, both in Poland and on the CME exchange, of nearly 100%. The prices in 2022 were among the highest over the period under review. In contrast, the price bottom coincides with lower prices in the wheat market. Similarly to the wheat market, the live cattle market in Poland has been less stable than the global market in the long term—the price volatility coefficients were 21.5% and 13.1%, respectively—but in the short term, the Polish market has been more stable than the global market—the standard deviation of the rates of return is 2.83 pp and 3.98 pp, respectively. The live cattle market is more stable than the wheat market, which can also be seen from the spread of returns, which ranges from −7.7% to 9.4% for the Polish market and −14.1% to 10.1% for the CME market.

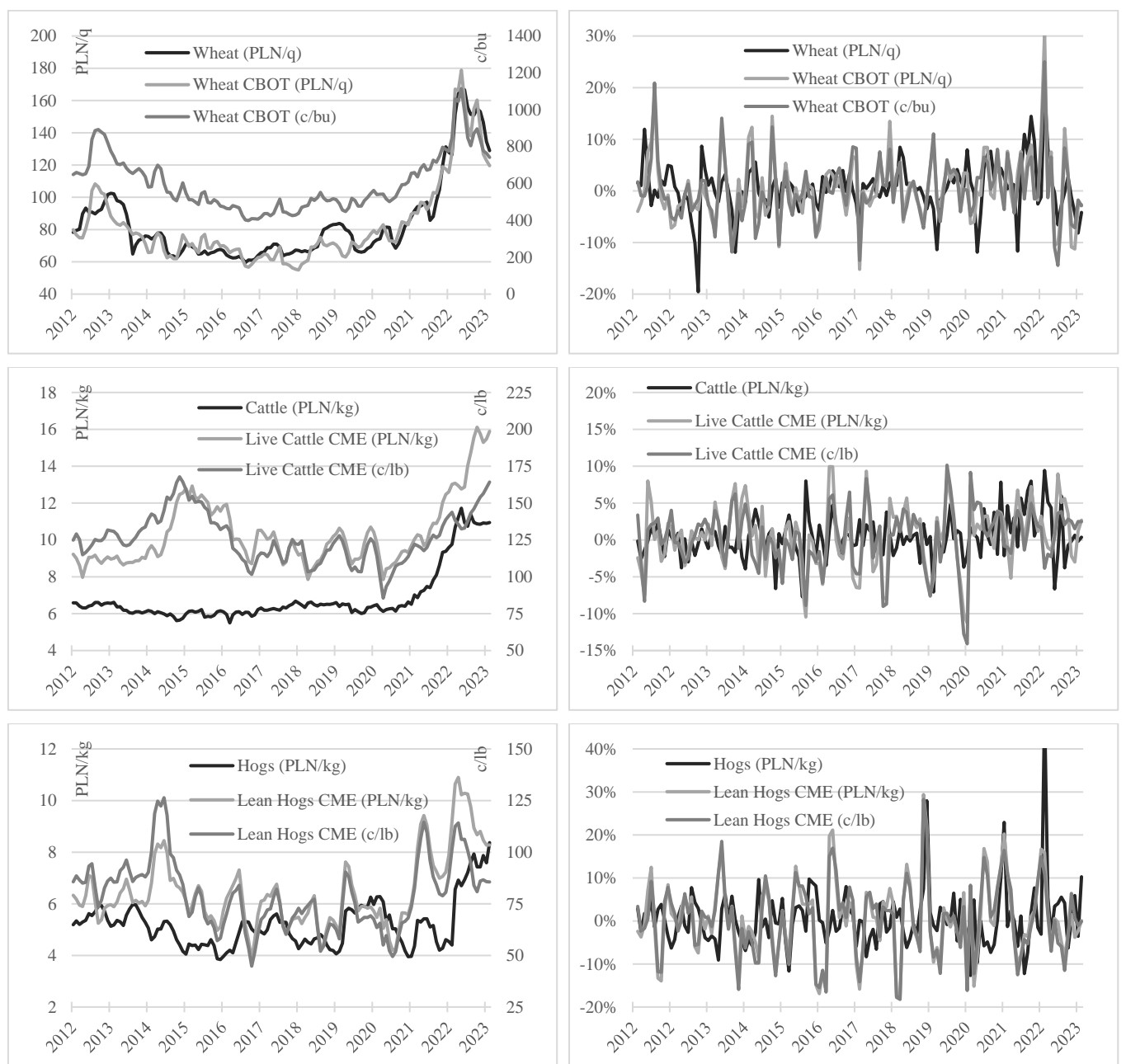

**Figure 1.** Price formation and rate of return of agricultural commodities.

**Table 1.** Descriptive time-series statistics of prices and rates of return for raw agricultural products.

| Raw Agricultural Products | Price Level | | | Rates of Return | | |
|---|---|---|---|---|---|---|
| | Mean | St.dev. | Vol.coef. | St.dev. | Min | Max |
| Wheat (PLN/q) | 84.01 | 25.71 | 30.6% | 4.99 | −19.5% | 19.9% |
| Wheat CBOT (c/bu) | 601.86 | 152.58 | 25.4% | 5.89 | −14.4% | 25.0% |
| Wheat CBOT (PLN/q) | 82.44 | 25.60 | 31.0% | 6.31 | −15.2% | 30.6% |
| Cattle (PLN/kg) | 6.85 | 1.47 | 21.5% | 2.83 | −7.7% | 9.4% |
| Live Cattle CME (c/lb) | 126.27 | 16.50 | 13.1% | 3.98 | −14.1% | 10.1% |
| Live Cattle CME (PLN/kg) | 10.34 | 1.83 | 17.7% | 4.28 | −11.3% | 9.9% |
| Hogs (PLN/kg) | 5.20 | 0.91 | 17.5% | 7.07 | −12.7% | 49.1% |
| Lean Hogs CME (c/lb) | 79.48 | 16.74 | 21.1% | 8.50 | −18.2% | 28.3% |
| Lean Hogs CME (PLN/kg) | 6.47 | 1.41 | 21.9% | 8.73 | −18.2% | 29.4% |

The prices of pig livestock (lean hogs) throughout the period were subject to quite significant changes. There were no sustained long-term trends here. However. the overall price formation was similar to that of wheat and live cattle, i.e., the lowest prices occurred in 2016–2021, and since 2021, there has been a noticeable increase in prices, which has reached 100% in Poland and the world. Long-term and short-term prices in Poland turned out to be more stable than in the world, with the price volatility coefficient being 17.5% and 21.1%, respectively, and the standard deviation of the rates of return being 7.07 pp and 8.50 pp. Poland saw the highest monthly change reaching 49.1% in March 2022. Generally, the long-term volatility of pig livestock prices does not differ significantly from the volatility of wheat and live cattle prices, while the short-term rates of return on pig livestock prices are significantly more volatile than the rate of return on wheat and live cattle. This is characteristic of the entire market, both Polish and global.

A rather interesting issue is the effect of the exchange rate on the volatility of prices and returns. In general, the conversion of prices from USD to PLN, i.e., world prices expressed in PLN, is characterized by higher volatility. Wheat saw an increase in the coefficient of price variation from 25.4% to 31.0%, live cattle saw an increase from 13.1% to 17.7% and for pig livestock there was an increase from 21.1% to 21.9%. In the case of rates of return, wheat saw an increase in the standard deviation of returns from 5.89 pp to 6.31 pp, live cattle saw an increase from 3.98 pp to 4.28 pp and pig livestock saw an increase from 8.50 pp to 8.73 pp.

The obtained results indicate the unfavorable impact of the exchange rate on the volatility of agricultural commodity quotations.

Agricultural commodity markets are generally characterized by low price stability. This is quite a big problem for agricultural producers, because it is difficult to plan revenues in such a situation, and thus to carry out investments. As research shows, the long-term risk of price volatility at the level of about 13–31% and short-term price volatility from −18% to 49% can cause serious budget perturbations.

### 3.2. Correlations of Prices and Rates of Return of Agricultural Raw Materials

Even a cursory observation of the development in prices of agricultural commodities (Figure 1) allows us to assume that prices in Poland and world prices are quite closely related. This is especially true for wheat, where Pearson's linear correlation coefficient of 0.8525 was obtained for the series of wheat prices in Poland and on the CBOT exchange, and it was as high as 0.9519 after adjusting for the exchange rate. For the rate of return series, the relationship is naturally weaker, but still very clear: 0.3397 in the original version, and 0.3862 after adjusting for the exchange rate.

Live cattle can be indicated as the second most-price-related product, where the level of relation between the original price series is 0.3318, and after adjusting for the exchange rate it is 0.7055. For rates of return, it is 0.1073 and 0.1516, respectively.

Weaker links apply to the pig livestock market, where the linkage of price series in the original version is 0.2955, and after taking into account the exchange rate it is 0.5085. For the return series, it is 0.1957 and 0.2323, respectively.

What is characteristic here are the positive values of the correlation coefficients, i.e., higher world prices correspond to higher prices in Poland, and lower prices in the world correspond to lower prices in Poland.

It is interesting to compare the values of Pearson's linear correlation coefficients and the values of Spearman's order coefficients. In general, in the case of rate of return series, there are no major differences here; so, we can talk about a weak linear dependence of rates of return. However, in the case of product prices in Poland and world prices, quite large spreads between these coefficients were obtained after adjusting for the exchange rate, especially for live cattle. However, the general conclusion regarding the positive correlation of agricultural prices is consistent.

It is worth noting that the high values of the Pearson and Spearman correlation coefficients and small differences between them for wheat prices allow us to consider this

market as subject to global influence. On the other hand, lower values of the Pearson and Spearman coefficients, as well as greater differences between them for the prices of live cattle and pork livestock, allow for recognizing these markets as markets that are subject to local rather than global conditions to a greater extent.

### 3.3. Causality in the Market of Agricultural Commodities

The correlation study makes it possible to determine the strength and direction of the relationship, and according to the results (Table 2), allows for the ranking of products from the most strongly related, i.e., wheat, to products with an average strong relationship, i.e., live cattle, to the least related products, i.e., pig livestock. This study, however, does not make it possible to establish a cause-and-effect relationship. The identification of a cause-and-effect relationship is an important issue from the point of view of the price-setting process, and also has a broader dimension, as it is related to the effectiveness of government intervention and protective measures of local agricultural markets.

**Table 2.** Correlation relationships of time series of prices and return rates of raw agricultural products.

| Raw Agricultural Products | Price Level (Non-Stationary Variables) | | | Rates of Return (Stationary Variables) | | |
|---|---|---|---|---|---|---|
| | r Pear. | R Spear. | m | r Pear. | R Spear. | m |
| Wheat (PLN/q) ↔ Wheat CBOT (c/bu) | 0.8525 | 0.8220 | 0.0511 | 0.3397 | 0.3126 | 0.0177 |
| Wheat (PLN/q) ↔ Wheat CBOT (PLN/q) | 0.9519 | 0.8472 | 0.1882 | 0.3862 | 0.3321 | 0.0388 |
| Cattle (PLN/kg) ↔ Live Cattle CME (c/lb) | 0.3318 | 0.0180 | 0.1098 | 0.1073 | 0.0912 | 0.0032 |
| Cattle (PLN/kg) ↔ Live Cattle CME (PLN/kg) | 0.7055 | 0.1680 | 0.4695 | 0.1516 | 0.1305 | 0.0060 |
| Hogs (PLN/kg) ↔ Lean Hogs CME (c/lb) | 0.2955 | 0.3306 | −0.0219 | 0.1957 | 0.0463 | 0.0361 |
| Hogs (PLN/kg) ↔ Lean Hogs CME (PLN/kg) | 0.5085 | 0.3107 | 0.1621 | 0.2323 | 0.0750 | 0.0484 |

Granger's causality study makes it possible to determine the cause and effect relationship in a statistical, not formal, sense. However, sometimes, such a statistical definition may be sufficient information in the field of shaping agricultural policy, as it is an expression of formal relationships that can be multidimensional and difficult to model. This statistical approach was used to examine the causal relationship in the market of wheat, live cattle and pig livestock (Table 3).

**Table 3.** Granger causality tests.

| Raw Agricultural Products | Rates of Return (Stationary Variables) | |
|---|---|---|
| | F Stat | p-Value |
| Wheat (PLN/q) → Wheat CBOT (c/bu) | 2.2094 | 0.1140 |
| Wheat CBOT (c/bu) → Wheat (PLN/q) | 4.2181 | 0.0169 |
| Cattle (PLN/kg) → Live Cattle CME (c/lb) | 0.0832 | 0.9202 |
| Live Cattle CME (c/lb) → Cattle (PLN/kg) | 0.7285 | 0.4846 |
| Hogs (PLN/kg) → Lean Hogs CME (c/lb) | 0.9813 | 0.3777 |
| Lean Hogs CME (c/lb) → Hogs (PLN/kg) | 5.0710 | 0.0076 |

$p$-Value < 0.05—statistically significant causality.

The research allows for the recognition of rates of return on the world wheat market as the reasons for shaping the rates of return in the Polish market; the respective level of significance was 0.0169. The causality in the opposite direction from the Polish wheat market to the world market was not found; the respective significance level was 0.1140.

The same conclusion applies to the rates of return for pig livestock. In this market, quite weak correlations were found, but the causality from the world market to the Polish

market turned out to be statistically significant, and the respective significance level was 0.0076. On the other hand, the Polish market for pig livestock cannot be considered as the causal factor for shaping the economic situation in the world market; the respective level of significance was 0.3777.

However, different results apply to the live cattle market. Here, no causality was found in either direction. For the causality from the world market to the Polish market, a significance level of 0.4846 was obtained, and the other way round from the Polish market to the world market was 0.9202.

For the wheat market (Table 4, rates of return), the following can be concluded:

- Poland's current wheat returns are strongly influenced by the 1-month lagged global wheat returns (*t*-Stat = 2.9043) and by their own lags (*t*-Stat = 3.5442).
- No significant impact of the rates of return for wheat in Poland on the rates of return for wheat in the world is clearly visible (*t*-Stat = −1.6529); global wheat returns are significantly influenced by their own lags (*t*-Stat = 4.5517).
- Comparing the values of the regression coefficients, the following can be seen:
  - For the wheat model in Poland, the regression coefficients of lagged 1-month returns are 0.3345 for Poland, and 0.2182 for the world; so, the coefficient for the world is slightly weaker than for Poland, and the same direction of influence is revealed;
  - For the wheat model in the world, the regression coefficients of the rates of return lagged by 1 month are −0.1887 for Poland, and 0.4137 for the world; so, the rates of return in the world do not clearly depend on Poland, but are subject to autocorrelation;
  - The significance of the returns lagged by 2 months is clearly weaker than for the returns lagged by 1 month, and their regression coefficients are mostly closer to 0.

**Table 4.** VAR model for wheat rates of return.

| Explanatory Variables | Rates of Return (Stationary Variables, Dependent Variables) | |
| --- | --- | --- |
| | Wheat (PLN/q) | Wheat CBOT (c/bu) |
| Wheat (PLN/q) (−1) | 0.3345 [3.5442] | −0.1887 [−1.6529] |
| Wheat (PLN/q) (−2) | −0.0507 [−0.5475] | 0.2019 [1.8008] |
| Wheat CBOT (c/bu) (−1) | 0.2182 [2.9043] | 0.4137 [4.5517] |
| Wheat CBOT (c/bu) (−2) | −0.0487 [−0.6423] | −0.2973 [−3.2377] |
| C | 0.0029 [0.7386] | 0.0022 [0.4562] |
| R-squared | 0.2109 | 0.1729 |

The table contains regression coefficients; *t*-Stat in [ ]; *t*-Stat > 2—statistical significance.

In the case of the VAR model for shaping the rates of return for live cattle (Table 5, rates of return), the following are concluded:

- The current rates of return for live cattle in Poland are independent of their lags (*t*-Stat = −0.0118) and of the lags of rates of return for live cattle worldwide (*t*-Stat = 1.0419);
- The current rates of return for live cattle in the world depend only on their lags (*t*-Stat = 6.4481); the lagged rates of return in Poland are not affected here (*t*-Stat = 0.1966);
- Comparing the values of the regression coefficients, the following can be seen:

- For the cattle model in Poland, the regression coefficients for all of the delays are very close to 0; so, the influence of the world on the rates of return in Poland is not revealed;
- For the global cattle model, the regression coefficients of return rates delayed by 1 month are 0.0213 for Poland and 0.5519 for the world; so, the rates of return in the world do not depend on Poland, but are subject to autocorrelation.

**Table 5.** VAR model for live cattle rates of return.

| Explanatory Variables | Rates of Return (Stationary Variables, Dependent Variables) | |
| --- | --- | --- |
| | Cattle (PLN/kg) | Live Cattle CME (c/lb) |
| Cattle (PLN/kg) (−1) | −0.0010 [−0.0118] | 0.0213 [0.1966] |
| Cattle (PLN/kg) (−2) | 0.0513 [0.5796] | 0.0387 [0.3569] |
| Live Cattle CME (c/lb) (−1) | 0.0728 [1.0419] | 0.5519 [6.4481] |
| Live Cattle CME (c/lb) (−2) | 0.0078 [0.1112] | −0.2669 [−3.1102] |
| C | 0.0040 [1.5867] | 0.0019 [0.6157] |
| R-squared | 0.0148 | 0.25117 |

The table contains regression coefficients; *t*-Stat in [ ]; *t*-Stat > 2—statistical significance.

In the case of the VAR model for shaping the rates of return for pig livestock (Table 6, rates of return), the following can be found:

- The current pig livestock returns in Poland are independent of their lags (*t*-Stat = 1.3232), but dependent on the lags of global returns (a *t*-Stat of 3.0280 was obtained); so, this is an unprecedented situation even for the wheat market, where the influence of both its lags and the world was revealed;
- The current global pig livestock returns are only dependent on their lags (*t*-Stat = 6.4894); there is no effect of lagged returns in Poland here (*t*-Stat = −0.1102);
- Comparing the values of the regression coefficients, the following can be seen:
  - For the pig livestock model in Poland, the regression coefficients of 1-month lagged returns in the world are about two times larger than for 1-month lagged returns in Poland, at 0.2391 and 0.1191, respectively; so, it is an unexpected situation that the world market returns dominate so strongly over domestic returns;
  - For the pig livestock model in the world, the regression coefficients of the rates of return lagged by 1 month are −0.0107 for Poland, and 0.5553 for the world; so, the rates of return in the world do not depend on Poland, but are subject to autocorrelation.

The results of the Granger causality test described in Table 3 are directly related to the relationships described in Tables 4–6 (VAR models).

It is worth noting that correlation studies and causality studies complement each other. Correlational studies have made it possible to determine the strength and direction of a relationship, while causality studies have made it possible to establish a cause-and-effect relationship. It is worth noting the following here:

- Wheat, which showed strong correlations, is subject to one-sided causal dependence from the world market to the Polish market;
- Live cattle, which showed average correlation relationships, did not show significant causal relationships;

- Pig livestock, which did not show very strong correlations, is subject, like wheat, to a one-way causal relationship from the world market to the Polish market.

**Table 6.** VAR model for pig livestock rates of return.

| Explanatory Variables | Rates of Return (Stationary Variables, Dependent Variables) | |
| --- | --- | --- |
| | Hogs (PLN/kg) | Lean Hogs CME (c/lb) |
| Hogs (PLN/kg) (−1) | 0.1191 [1.3232] | −0.0107 [−0.1102] |
| Hogs (PLN/kg) (−2) | −0.0267 [−0.3044] | −0.1306 [−1.3726] |
| Lean Hogs CME (c/lb) (−1) | 0.2391 [3.0280] | 0.5553 [6.4894] |
| Lean Hogs CME (c/lb) (−2) | −0.0307 [−0.3752] | −0.2462 [−2.7695] |
| C | 0.0047 [0.7871] | 0.0033 [0.5135] |
| R-squared | 0.0994 | 0.2690 |

The table contains regression coefficients; *t*-Stat in [ ]; *t*-Stat > 2—statistical significance.

## 4. Discussion

Shaping the prices of agricultural commodities is not a simple matter. The number of factors affecting prices is so large that it is very difficult to make reliable forecasts, even for a short one-year period [52,53]. This causes periodic tensions between agricultural producers and the government. Often, in the situation of low purchase prices of agricultural raw materials, agricultural producers start protest actions, and their demands concern the increase in the profitability of agricultural production [54]. Meanwhile, the whole system of the European Union market related to agricultural policy is the system through which the most money passes; no other action, such as aid to agricultural producers, is so costly [55,56]. Nevertheless, the EU market, and in particular the Polish market, is still a price-volatile market.

The expectations of agricultural producers in terms of ensuring the profitability of agricultural production can, of course, be considered to be justified, especially if we take into account the length of the production cycle, which is one year for grains, over a year for live cattle and several months for pig livestock. Sometimes, poor yields with low purchase prices can cause months of difficulties for agricultural producers [57,58]. Additionally, this often happens for reasons beyond the control of agricultural producers; sometimes, it is hard to even expect help from purchasing companies, and it is hard to expect CSR [59]. On the other hand, according to economic theories, the condition for stable and sustainable economic growth is the lack of trade barriers between countries. Meanwhile, the state interventionism expected by agricultural entrepreneurs is in contradiction with the principles of the free market. In addition, to ensure the effectiveness of intervention activities, it is necessary to protect the market against the influx of raw materials from abroad [60,61]. Therefore, it is crucial to determine whether a given domestic market is subject to global influence and is integrated with the world, or whether it is a market isolated from the global economic situation.

The results of the research conducted in this article for Poland allow us to unambiguously recognize the wheat market as a price-integrated market with the world market. This is strongly evidenced by the correlation relationship and the fact that the Polish market follows changes in world prices. Furthermore, prices in Poland are at a similar level to world prices. So, this is an example of a product for which the law of one price works absolutely perfectly. This has both good and bad sides. For the general public, it means

greater food security, but for agricultural producers, it means that they have to reckon with foreign competition. It also limits the effectiveness of the state's intervention policy. Studies have shown a lag of about one month between prices in Poland and changes in world prices, which allows for short-term price forecasting, which can be used for commercial decisions to a certain limited extent. The government's yielding to pressure from wheat producers in Poland will be costly for the state budget and ineffective in the long term.

The second product studied was live cattle. Here, integration with the global market is average and consists of an average strong correlation. However, a causal relationship could not be established. The reason why it was not possible to determine the causal relationship may be due to the reaction of prices in Poland to changes in world prices shorter than 1 month, but also to the fact that the transport of meat is not as easy to implement as the transport of grains. So, naturally, the links in this market will be weaker. To some extent, it can be used for market intervention, especially since the purchase prices of live cattle in Poland are clearly lower than world prices.

The third product was pig livestock. Here, despite weaker correlation links, it was found that the prices of this commodity in Poland are influenced by world prices; so the market is integrated, albeit weakly, with the world market. As in the case of the prices of live cattle, the purchase prices of pig livestock in Poland are at a lower level than world prices. This provides a reserve for intervention in this market. It is worth noting that Polish agriculture is quite fragmented, i.e., there are a relatively large number of small farms in Poland, which gives buying agents an advantage over agricultural producers, who have no bargaining power. A certain practical solution could be the association of farmers, so that their bargaining power would be greater than that of purchasers. Such processes are promoted by the government and the European Union, and research shows that this direction is right.

Referring to the questions posed in the paper, it is clear that the Polish agricultural commodity market is a recipient of world prices. It is a small market on a global scale and is subject to global processes. The Polish market is not able to shake the world market, while the economic situation in the world market quite easily transfers to the domestic market. Different volatility indices give a different classification of the level of price volatility and rates of return in Poland against the background of global volatility. Price volatility from world markets can be organic by the effect of the exchange rate, as is often the case, for example, in the case of energy commodities [62,63], but in the case of agricultural markets, the exchange rate reinforces volatility. Such results are not favorable for the domestic economy and may further increase the dissatisfaction of domestic agricultural producers [64,65].

Research allows for the considering of agricultural markets as world markets. The globalization processes taking place on them cause prices from one market to move very quickly to other markets [47,66–68]. This process is all the more rapid and integrated when the products are stronger and easier to transport. Wheat is one of the commodities that is easily transported, whereas live cattle and pig livestock are more difficult to transport; therefore, the wheat market turns out to be a market with a very high degree of price integration, and the markets for live cattle and pig livestock are less integrated.

**Funding:** This research received no external funding.

**Institutional Review Board Statement:** Not applicable.

**Data Availability Statement:** Data available to the public in relevant institutions.

**Conflicts of Interest:** The author declares no conflict of interest.

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
