# Peer review of "Are Small Agricultural Markets Recipients of World Prices? The Case of Poland"

_agriculture, doi:10.3390/agriculture13061214_

Round 1

Reviewer 1 Report

Are small agricultural markets recipients of world prices? 2 A case for Poland

This study assesses if Polish agricultural market is a recipient of prices from the world market and if the price volatility of the domestic market is higher or lower than the price volatility of world markets. The author adopts a VAR methodology for the years 2012-2022. The results show that the market of wheat is the most strongly price-integrated, while the market of beef and pork livestock show a slightly weaker integration.

The study investigates an interesting theme that deserves attention. However, the contributions to the literature, some technical aspects, the discussion, and the importance of the findings should be strongly improved. Specific comments are given in what follows.

Major Comments

Abstract

The authors should pay attention to the use of language throughout the paper.

Introduction

·         Overall, the introduction needs to improve. The author should better motivate their study by explaining why it is necessary, and what policy-level problem is addressing. The author should elaborate on the contributions to the existing literature and should evidence the gap in research.

·         Line 33-34. Delete “that food imports be blocked” and write “for protective tariffs”.

·         Line 35-36 should be deleted, i.e. “Meanwhile, the common European market 35 contradicts the expectations of domestic agricultural producers”.

·         Line 39. Cite some studies on transmission in addition to [2], namely the analysis by Pierre and Kaminski (2019).  

·         A part of the literature linked to financialization in commodity markets should be also considered. Line 97-98. Please modify the sentence “Most common among these other factors is the so-called financialization of forward agricultural markets” in “Most common among these other factors is the so-called financialization of agricultural commodity markets [e.g. Algieri et al., 2017; Algieri, 2016]”

·         Line 103-104. Please modify the sentence “…as an asset class provide a hedge against positions in capital markets” in “…as an asset class provide a hedge against other securities in financial markets.

·         Line 138. Use the term “biofuels” instead of “biocomponents”.

·         Line 195. Please use the term “trade” instead of “exchange”.

·         Line 203. The term “liquidate” is not correct in that context. Rewrite the sentence

Methodology

·         The part on Granger Causality should be rewritten. From line 255 to 270. Please refer to an econometric book.

·         The major problem of this study is that the Granger causality and VAR can be conducted on stationary series. The author should consider therefore only returns and not prices, since price are not stationary. The author should proceed in steps: carry out the unit root tests and then Granger and VAR.

Economic discussion

·         More elaboration on the economic implications of the findings and why are the results sensitive to the type of agricultural commodity

References:

Algieri, B. 2016. Conditional price volatility, speculation, and excessive speculation in commodity markets: sheep or shepherd behaviour?, International Review of Applied Economics, 30(2): 210-237

Pierre G., Kaminski J. (2019) Cross country maize market linkages in Africa: integration and price transmission across local and global markets, Agricultural Economics, 50: 79-90

Algieri B, Kalkuhl M, Koch N, 2017. A tale of two tails: Explaining extreme events in financialized agricultural markets, Food Policy 69: 256-269

Edit your English

Reviewer 2 Report

Comments on “Are small agriculural markets recipients of world prices? A case for Poland”

This paper uses time series econometric methods to test whether commodity prices in Poland are Granger caused by world prices as defined by prices on the Chicago Board of Trade and the Mercantile Exchange. The time frame is monthly average prices from 2012 – 2022.

The paper also includes a graphical presentation of the data which is useful for defining the data sets. It also includes a report of correlation between Chicago and Poland prices which adds no value.  It is not the case that the correlation study makes it possible to determine the direction of the relationship – there is no causal inference possible.

The author needs to explain why such a short time series will allow valid inference. The short time frame is further complicated by the fact that three of the years are during the pandemic when trade was disrupted and the time series model likely experienced structural change.

It was not clear why Poland is an interesting case. The paper mentions exchange rates, but the zloty trades very narrowly from 0.21 – 0.25 euro per zloty over the sample period, so the currency valuations are quite stable. Why would the Polish case be any different from other European countries?

I could see that it would be interesting to compare the degree of integration between Polish and world markets before and after transition or before and after the pandemic, but that is not done.

The paper is sloppy in its applications of standard econometric tests. First, are these time series stationary? I would presume that commodity prices would be random walks, even if averaged over a month. If these series have unit roots, do the VARs need to be in differenced form to yield stationary series? Doesn’t that affect the Granger causality tests?

I would think that the Polish and world prices would be subject to some sort of arbitrage condition that would yield stable equilibrium relationships. That would mean that the Polish and World price series would be cointegrated. Standard VAR would not apply. Johansen tests would likely find that the series are cointegrated which would be useful information for the author’s hypotheses that the Polish prices reflect world prices. We might even find that the cointegration breaks down in the pandemic.

English is fine

Round 2

Reviewer 1 Report

The author has revised the paper along the suggested lines. The revision has helped the author to refine her contribution to the literature and to better reframe the empirical analysis. However, I have some further indications to address and some technical parts to change before publishing:

  • Line 37. Change “protective tariffs” in “protective measures, including tariffs and bans”.
  • Line 139. Change “The compulsory use of biofuels in fuels has resulted” in  “The use of biofuels has resulted”
  • Line 227. Change “The empirical material studied is” in “The considered data for the empirical analysis is”
  • Lines 232-233 and 239-240. To be precise add the definitions of q kg c bu lb.
  • Table 3 Granger Causality Tests. Delete the columns concerning Price levels, since Granger CANNOT be carried out on non-stationary series. THIS IS VERY IMPORTANT
  • Tables 4-5-6 VAR model. Delete the columns concerning Price levels, since VAR CANNOT be carried out on non-stationary series. THIS IS VERY IMPORTANT
  • Adjust the comments.

Minor editing of English language required

Reviewer 2 Report

The paper is improved.

I am ok with acceptance
